# Posturographic Analysis in Patients Affected by Central and Peripheral Visual Impairment

**DOI:** 10.3390/jpm12101709

**Published:** 2022-10-13

**Authors:** Gabriella Cadoni, Pasqualina Maria Picciotti, Rolando Rolesi, Marco Sulfaro, Margherita Guidobaldi, Filippo Amore, Guido Conti, Gaetano Paludetti, Simona Turco

**Affiliations:** 1Fondazione Policlinico Universitario Agostino Gemelli IRCCS, Institute of Otolaryngology, Università Cattolica Sacro Cuore, 00168 Rome, Italy; 2National Center for Services and Research for the Prevention of Blindness and Rehabilitation of the Visually Impaired, Fondazione Policlinico Universitario Agostino Gemelli IRCCS, 00168 Rome, Italy

**Keywords:** central low vision, peripheral low vision, postural control, vestibular evaluation, computed dynamic posturography, motor control test

## Abstract

Although vision loss is known to affect equilibrium maintenance, postural control in patients affected by low vision has been poorly investigated. We evaluated postural stability and the ability to use visual, proprioceptive and vestibular information in different low vision patterns. Ten adults with normal vision (NC), fourteen adults affected by central visual impairment (CLV) and eight adults affected by peripheral visual impairment (PLV) were enrolled in our study. Patients underwent visual, vestibular and postural evaluation (bedside examination, Computed Dynamic Posturograophy). Motor Control Tests were performed to analyze automatic postural adaptive responses elicited by unexpected postural disturbances. Clinical evaluations did not show abnormality in all patients. In the Sensory Organization Test, CLV and PLV patients performed more poorly in conditions 3–6 and 3–4, as compared to NC subjects. The condition 5 score was significantly lower in the CLV group with respect to the PLV patients. Composite equilibrium scores demonstrated significant differences between low-vision subjects vs. NC subjects. No differences were found for somatosensorial contribution. Visual afferences showed lower values in all visually impaired subjects, while vestibular contribution was lower in the CLV patients as compared to the NC and PLV patients. MCT latencies were significantly worse in the CLV subjects. In the low-vision patients, postural control was modified with a specific pattern of strategy adaptation. Different modulations of postural control and different adaptive responses seemed to characterize CLV patients as compared to PLV subjects.

## 1. Introduction

Sensory information from somatosensory, vestibular and visual systems are integrated to provide equilibrium maintenance. Different studies focused on postural control have investigated how sensory inputs are reweighted, and how neural strategies change in different situations to control balance and postural reactions to perturbations [1]. The vestibular system acts by tracking the position and the movements of the head, while the visual system gives a spatial estimate of the position of objects relative to the body; finally, the proprioceptive system monitors the relative positions of different parts of the body [2].

Low Vision (LV) is a bilateral, severe and irreversible degree of vision loss that cannot be corrected by medical or surgical treatments, or with conventional eyeglasses. In according to the WHO, LV is the condition in which visual acuity is less than 6/18 and equal to or better than 3/60 in the better eye with best correction.

This condition severely affects the individual’s ability to socialize, read or drive, which affects their perceived quality of life [3]. It is strongly associated with older people, due to age-related eye diseases. Some of the most common causes include macular degeneration, glaucoma, diabetic retinopathy and retinitis pigmentosa. There are three different clinical patterns: central, peripheral and generalized types.

Central LV is characterized by a significant reduction in central visual acuity for near/far vision. Patients report a reduction in contrast sensitivity, facial recognition, stereopsis and reading skills [4]. Related diseases are represented by maculopathy, congenital and early-onset (Stargardt disease and chronic dystrophies), or are age-related, such as macular degeneration and pathological myopia. The trend of maculopathy leads to a progressive impairment in visual acuity eventually related to the development of a central scotoma, due to photoreceptor degeneration. Dense scotoma is a localized visual field defect that creates a blind spot, while in a relative scotoma there is a visual depression, but not a complete loss in light perception. In these patients, an absolute central scotoma, due to the loss of foveal fixation, leads to the development of a retinal area of eccentric fixation, the so called Preferred Retinal Locus (PRL). In chronic macular damage, the process is usually progressive and steady over time; in acute cases, the PRL is unstable and may have multiple and variable localizations [5].

In the peripheral LV, the central vision may be preserved, even though the wide-angle field of vision is affected by the extra-foveal retina or the optic nerve involvement, due to glaucoma [6] and retinitis pigmentosa [7]. As a consequence of progressive peripheral visual field impairment, patients lose the ability to move independently, especially in low-light conditions.

Generalized vision loss involves both the central and peripheral vision. This condition can be related to different diseases. Above all, diabetic retinopathy is associated with maculopathy, advanced retinitis pigmentosa and pathologic myopia. Patients show a disability in conditions requiring fine detail, such as reading and driving, as well as in outdoor mobility.

The loss of vision is known to affect postural control in blind subjects. Postural control in blindness has been investigated, and an interesting review suggests that improved remaining sensations in the presence of adaptations and neuroplasticity did not translate into better postural control performance [8]. However, specific postural control patterns in these patients were not quite clarified. Postural control in patients affected by low vision has been poorly investigated, and a few results suggest that, without visual references, patients employed adaptive changes by developing different sensory-motor interactions for gait and posture [9,10].

In this study, we evaluated balance in low-vision patients using Computerized Dynamic posturography (CDP), which is designed to quantitatively assess an individual’s ability to use visual, proprioceptive and vestibular cues to maintain postural stability. The goal of our research was to compare patients with different clinical patterns of vision impairment to clarify adaptation and plasticity conditions.

## 2. Materials and Methods

The study was approved by the local ethics committee (Prot. A001); participants gave written informed consent prior to participating. We enrolled 32 patients: 10 patients (5 F, 5 M; mean age 57 ± 11.2 SD) with normal vision and no vestibular impairment (control group, NC) and 22 (9 F, 13 M; mean age 59 ± 12.7 SD) with a minimum 5-year history of low vision from the “National Center of Services and Research of Blindness and Rehabilitation of the Visually Impaired” at the Agostino Gemelli Policlinic Foundation University, a Scientific Institute for Research, Hospitalization and Healthcare. When accounting for Neurocom international normative data on Equitest Equilibrium Scores in healthy adult populations, and considering an expected result sufficient to obtain a pathological Equilibrium Score, our sample size, although limited, was adequate (study power of 95%, Zpwr value = 1.64, *p*-value = 0.01, significance = 99% Zcrit value = 2.58). Low-vision patients were affected by visual impairment of both central (14 subjects, group CLV) and peripheral (8 subjects, group PLV) types. CLV patients had a Best Corrected Visual Acuity (BCVA) between 0.6 logMAR and No Light Perception (NLP). PLV patients had Peripheral Binocular Visual Fields of less than 30%. Enrolled patients had never undergone rehabilitation protocols.

### 2.1. Visual Evaluation

We performed the standard protocol of clinical assessment:Best Corrected Visual Acuity (BCVA): assessed through Early Treatment Diabetic Retinopathy Study charts. BCVA was expressed in logMAR values at a distance of 4 m with the optimal refractive correction.Reading Acuity (RA): assessed by the Minnesota Reading test (MNRead) charts at 25 cm using +4.00 sph (1×) reading lenses, in addition to the refractive adjustment distance.Contrast sensitivity: evaluated through Pelli Robson charts at a distance of 1 m, with +1.00 sph lenses, in addition to the refractive adjustment distance.Fixation stability was assessed using the Nidek Technologies MP-1 microperimeter. Patients focused on a central target for 30 s. The target shown was a white cross with an arm extension of 1°, but it was increased to ≥2° if patients were not able to see it. The fixation stability was classified according to Fujii et al.’s and Sawa et al.’s criteria.Retinal threshold sensitivity was assessed with the MP-1 microperimeter (Nidek Technologies, Albignasego, Italy) in manual mode use, using a specific 4–2 threshold strategy and the Goldmann III stimulus. The microperimetric examination was performed using the same number of stimulus spots for the scotoma area and residual area surrounding the PRL (Preferred Retinal Locus). Points were arranged similarly to an automatic pattern of perimetric examination. Overall, this strategy was associated with a lower procedure time and an optimal definition of the near-PRL vision area.Peripheral Binocular Visual Field (BVF): evaluated with a Humphrey Field Analyzer II (Carl Zeiss Meditec AG, Jena, Germany) automated static perimeter, based on three threshold stimuli programs that analyzed 100 points—36 points in the peripheral visual field, and 64 in the central part. This examination allowed for an optimal evaluation of the functional visual field. It took into account the most important functional perimetric areas (the paracentral and inferior visual field), and increased the number of points explored in those areas, whose integrity was fundamental to ensure autonomy in the environment.

### 2.2. Vestibular Evaluation

Vestibular function was clinically evaluated by bedside examinations consisting of: spontaneous nystagmus research; Fukuda tests; star-shaped march tests and index finger tests (for vestibulo-spinal examination); OTRs; skew deviations; ocular torsions (counterrolling) and head tilts (otolithic signs); clinical Head Thrusts or Impulse Tests (Halmagyi); Head Shaking Tests and Provocative maneuvers (the positional test, vibratory test and fistula test).

### 2.3. Postural Evaluation

Postural control was evaluated using Computed Dynamic Posturography (CDP), performed by Equitest, Neurocom Int. Inc., in Clackamas, OR, USA. The CDP was based on the Sensory Organization Test (SOT), which evaluates the contribution of different sensorial afferences and the Motor Control Test (MCT) to evaluate the effectiveness of the patient’s automatic motor reflex responses to restore balance following unexpected perturbations. 

In the SOT, the subject stood on a double forceplate enclosed by a visual surrounding. The dual forceplate recorded the vertical forces between the feet and the ground, as well as the horizontal shear forces, thereby allowing estimation of the position of the swaying body. Six different test conditions, each lasting 20 s, were repeated three times to obtain more reliable values (Figure 1A).

The Equilibrium Score (ES) was calculated for each condition and indicated the range of oscillation of the angle with respect to the vertical earth. It was calculated for each condition using the following formula: ES = ([12.5° − (THETAmax-THETAmin)] × 100)/12.5° (THETA was the angle between a line extending vertically from the center of foot support and a line extending from the center of foot support through the center of gravity.)

The data obtained also included the Composite Equilibrium Score (CES), which is a weighted average of the six conditions. The Sensory Analysis (SA) showed the contribution provided by the different sensorial afferences, and was obtained by the ratio of the different conditions (Figure 1B).

The Motor Control Test (MCT) challenged postural control by creating unexpected postural disturbances of different sizes (small, medium and large) through backwards and forwards platform translations. These translations elicited automatic postural responses. Parameters analyzed were:(1)Weight Symmetry (WS) (the relative distribution of weight on each leg);(2)Latency (the time between translation onset and the active response of the patient’s leg);(3)Composite Latency Score (CLS) (the average of the individual scores, considering both legs);(4)Strength Symmetry (the amplitude scale for the legs and three translations).

Data obtained from the patients were compared with those of the age-matched NC group subjects. We analyzed data obtained from the SOT and SA as a mean ± SD for the different conditions (1–6), CES and sensorial afferences (somatosensorial, visual, vestibular and visual preference). For the MCT data, we considered the mean values of latency for each leg, in backwards and forwards translations of medium and large sizes. The results were presented as means ± the standard error of the mean (SEM), and differences were assessed by using variance analysis ANOVA (Statistica, Statsoft, Tulsa, OK, USA); a *p* value < 0.05 was considered significant.

## 3. Results

### 3.1. Ophthalmologic Analyses

Ophthalmologic assessment is reported in Table 1 and Table 2. In the CLV patients, the mean BCVA for the best eye was 0.9 (±0.1 SD) logMAR and 1.0 (±0.1 SD) logMAR for the worst eye; the mean of the PRL locations for the best eye was 7.3 (±4.2 SD) degrees, while that of the worst eye was 10 (±6.2 SD) degrees, compared to the fovea (Table 1, Figure 2A).

In the PLV patients, the mean BCVA for the best eye was 0.5 (±0.3 SD) logMAR and 0.6 (±0.6 SD) logMAR for the worst eye; the mean BVF was 12.2% (±7.7 SD) (Table 2, Figure 2B).

### 3.2. Vestibular Analyses

Vestibular evaluations did not show abnormality. Spontaneous and/or positional nystagmus was not observed; otholitic signs (OTR, skew deviation, ocular torsion and head tilt) were absent, and the Impulse Tests, Head Shaking Tests, Vibratory and Fistula tests were negative. Finally, the clinical vestibulo-spinal examinations did not identify alteration.

Table 3 shows data obtained from the SOT, SA and MCT. No significant differences were observed between the LV patients and the normal control in conditions 1–2. The CLV patients performed more poorly in conditions 3 (*p* < 0.01) and 4–6 (*p* < 0.001), with significant differences from the NC in all four tests, and from the PLV for condition 5 (Figure 3A, *p* < 0.001). Data from the PLV patients were significantly different from the NC for conditions 3 (*p* < 0.01) and 4 (*p* < 0.001). Analysis of the CES demonstrated significant differences between both the central and the peripheral LV patients vs. the normal controls (Figure 3A, *p* < 0.001).

The Sensory Analysis showed the worst results in the CLV patients. Specifically, the comparison of vestibular data demonstrated significant differences between this group and both the NC and the PLV, while analysis of the visual preference situation detected a significant difference only between the CLV and the NC groups. Regarding the visual component, it was quite obvious that visual afferences showed significantly higher values in the NC compared to the LV patients, without differences between the central and peripherical LV patients. No differences were found for the somatosensorial contribution (Figure 3B).

The lower section of Table 3 shows data obtained from the MCT sequences of medium/large translations in the backward and forward directions. The latency values of the CLV patients were significantly worse when compared to the NC and the PLV (Figure 4). No statistical differences existed between the NC and PLV groups, except for the medium forward and backward translations on the left leg (Table 3).

## 4. Discussion

The visual system plays a major role in postural control, and postural sway increases with the absence/impairment of vision. Our study showed that, in low-vision patients, postural control was modified with a specific pattern of strategy adaptation. Moreover, we found that postural stability was different between central-low-vision and peripheral patients with a different reorganization. To our knowledge, this is the first report investigating the differences between central and peripheral low-vision patients.

Sensory organization tests can identify modifications in the use of the somatosensory, visual and vestibular systems, by providing the eyes, feet and joints with inaccurate information. Whenever a subject experiences a conflict in one or more senses, an adaptive response occurs, in which the individual suppresses or ignores the responses from those senses and selects more accurate sensory systems to generate the appropriate motor response.

In our patients, we found that patients affected by low vision had worse Composite Equilibrium Scores and Equilibrium Scores in conditions 3–6. However, static balance was almost normal in the LV patients. Our results are in agreement with Tomomitsu et al. [11], who demonstrated that low-vision individuals had worse postural stability than normal-vision adults, in terms of dynamic tests and balance on foam surfaces [11].

Postural control is certainly a complex process that can be influenced by many factors, such as aging, physical conditions, cognitive functions and many age-related diseases. Particularly in elderly populations, cognitive or physical inactivity could be associated with lower balance control, leading to a higher risk of falling. The sample size of our study was not adequate for a stratification of the data relating to these variables. To obtain as homogeneous a sample as possible, our study protocol excluded patients who underwent any kind of major surgery, or suffered from serious pathologies (neurological, orthopedic, physiatric, vestibular, otologic or other serious comorbidities). We enrolled subjects with healthy lifestyles, adequate personal autonomy and age-appropriate physical activity, as many authors underlined that even leisure physical activity was very effective for balance control amelioration and fall prevention.

Several studies have suggested that vision impairment can increase postural instability and interaction between the central nervous, muscular and peripheral sensory systems, which are fundamental for calibrating sensory maps and adjusting balance [12,13]. Our study suggested that when a subject was given erroneous visual cues, as in our series, the visually dominant (visual preference) subjects could not suppress inaccurate information, which means balance was disrupted.

Patient comparisons showed that the CLV group had the worst performance, which was significantly different for condition 5 of the SOT. In the literature, there are contradictory findings concerning the respective contributions of the two visual systems in postural control [14]. It was demonstrated by Brandt et al. [15] that the central and peripheral retina had different involvements in postural stabilization, as the first one dominated pattern recognition and motion detection, while the peripheral/paracentral retina was largely involved in visually-induced vection. In agreement with our data on low-vision patients, Straube et al. [16,17] demonstrated that body sway applied to a force-measuring platform was lower for low-central-vision subjects than for normal subjects. Our results were about how Sensory Analysis demonstrated a new hierarchy and sensory reorganization. Sensory Analysis highlighted that if, there was an obvious reduction in the visual component in all affected patients, the vestibular component/visual preference was only significantly impaired in the CLV group. The influence of the visual system on postural control has been documented in several studies, especially in individuals with low vision [8]. Patients with visual dysfunction must place a greater demand on somatosensory and vestibular information to maintain postural stability, establish and connect movement patterns and adjust to positions in space to compensate for low-functioning visual systems. Studies comparing blind and seeing individuals in static- and dynamic- balance tasks confirmed that ≈80% of an individual’s sensory perception was gathered by the visual system [18], which processed/integrated other sensory inputs to select a balancing strategy [13]. However, currently, few studies have examined postural stability by comparing low- vs. normal-vision adults, and no studies have compared different types of LV. We justified the impairment of the vestibular component and visual preference in central-low-vision patients with the role of the central and peripheral vision in the control of posture [14] and different cross-modal plasticity compensations. We hypothesize that, in the CLV group, peripheral vision may have affected postural control through a visuo-vestibular conflict. Visual and vestibular inputs may have interacted through a ‘reciprocal inhibition’, whereby both systems competed to suppress the other, in order to produce a coherent sense of self motion [19]. This suggestion could justify the impairment of both the vestibular and preferential components in the SA for the CLV patients.

The MCT latency values of the CLV patients were significantly worse compared to the NC and the PLV groups. Under these conditions, central low vision adversely affected adaptive responses. This result could have depended on different adaptations occurring to increase kinesthetic information that compensated for unreliable/incomplete visual information.

Moreover, we also demonstrated differences between the NC and PLV groups for translations on the left leg. This finding could be related to a different distribution of weight on the legs, due to the visual deficits. Very few studies have examined the postural control of low-vision subjects in a single-leg stance; however, Tomomitsu et al. [11] analyzed the Unilateral Stance, and demonstrated a difference in the left leg test in low-vision patients vs. the NC. As previous studies suggested, the proprioception inputs could have been overloading the left leg, and unilateral stance tasks might have depended on some neuromuscular requirement and muscular strength [12,20,21]. The ability of the postural control system to select a higher joint configuration variance may have contributed to the maintenance of postural stability by correcting lower extremity movements in the individuals with vision impairments [11,22].

## 5. Conclusions

Although this study is preliminary, our data showed that patients affected by low vision had a postural control modification, and the detected impairment was greater in the presence of central low vision. This finding is very interesting, on the one hand, for clinical practice, since our preliminary results suggested the need for rehabilitative interventions to reduce the risks of falling in this population, and, on the other hand, for the implementation of possibly more personalized perspectives on rehabilitation.

## Figures and Tables

**Figure 1 jpm-12-01709-f001:**
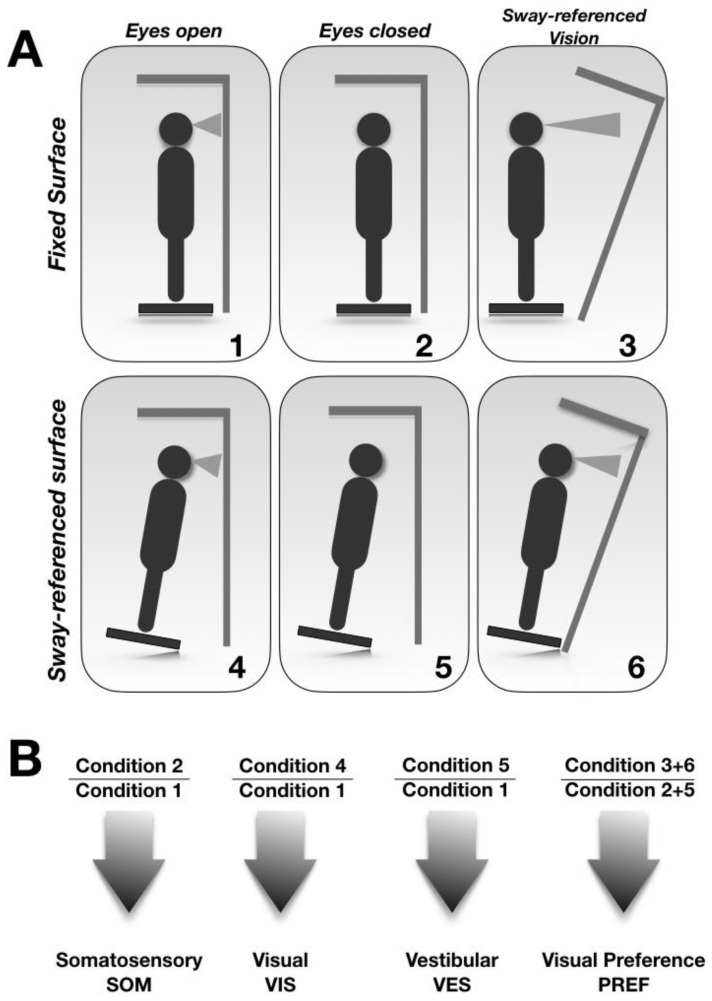
Computed Dynamic Posturography (see text for details). Six different test conditions, each lasting 20 s, were repeated three times to obtain more reliable values (**A**). The Sensory Analysis (SA) showed the contribution provided by the different sensorial afferences, and was obtained by the ratio of the different conditions (**B**).

**Figure 2 jpm-12-01709-f002:**
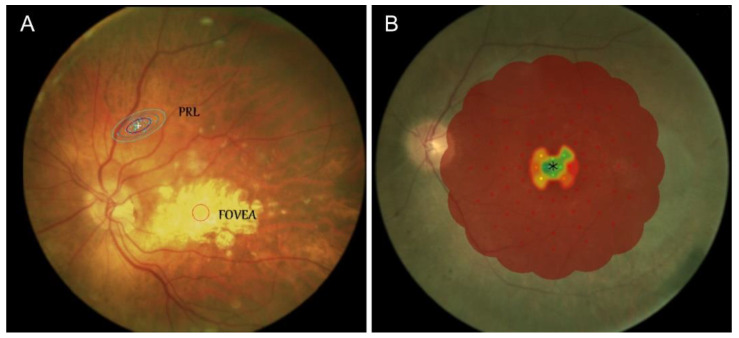
Preferred Retinal Locus (PRL) in visually impaired patients: (**A**) PRL location in patients with central low vision; (**B**) Microperimetric examination (MP-1) of a patient with severe peripheral vision loss. The green central points represent the central-vision area preserved, while the large red area identifies irreversible retinal damage.

**Figure 3 jpm-12-01709-f003:**
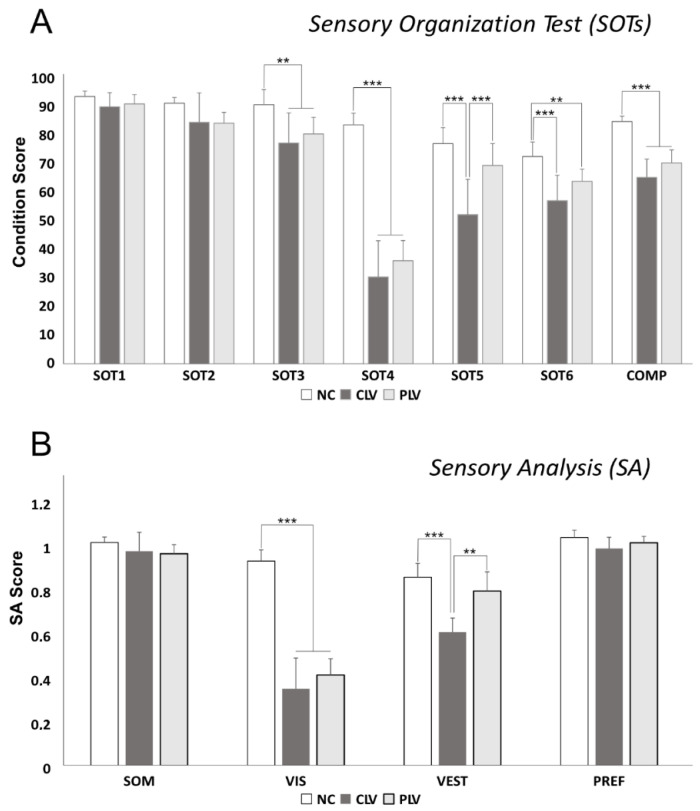
Sensory Organization Test (SOT) and Sensory Analysis (SA) scores: (**A**) Graphs show values for each SOT condition (SOT1–6) and composite scores (COMP). NC: Normal Control; CLV: central low vision PLV: peripheral vision loss. (** *p* < 0.01; *** *p* < 0.0001). (**B**) Graphs show Sensory Analysis (SA) scores. SOM: somatosensorial contribution; VIS: visual component VEST: vestibular component. NC: Normal Control; CLV: central low vision PLV: peripheral vision loss. ** *p* < 0.01; *** *p* < 0.0001.

**Figure 4 jpm-12-01709-f004:**
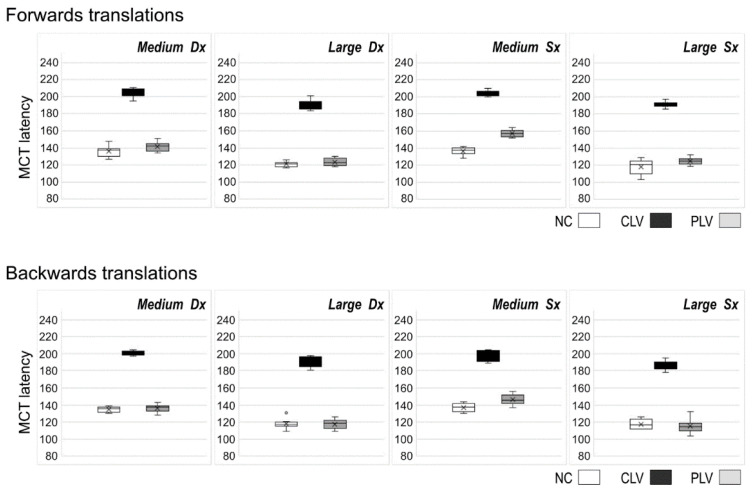
Mean MCT amplitudes for the medium/large forward and backward linear-translations. NC: Normal Control; CLV: central low vision PLV: peripheral vision loss.

**Table 1 jpm-12-01709-t001:** Patients with central vision loss characteristics.

ID	Age	Gender	Eye Disease	BCVA (LogMAR) Best Eye	BCVA (LogMAR) Worst Eye	PRL Best Eye	PRL Worst Eye
P1	49	M	Pathologic myopia	1.3	HM	12°	
P2	48	W	Stargardt disease	0.9	1	6°	8°
P3	82	W	AMD	1	1	16°	18°
P4	43	M	Stargardt disease	1.3	1.3	6°	8°
P5	71	W	Pathologic myopia	0.7	HM	6°	
P6	70	M	Pathologic myopia	0.9	1	6°	6°
P7	68	M	Pathologic myopia	0.8	CF	8°	
P8	46	W	Stargardt disease	1	1	10°	12°
P9	52	W	Stargardt disease	0.9	1	14°	20°
P10	58	M	Pathologic myopia	0.8	1	2°	2°
P11	38	M	Pathologic myopia	1	PROSTHESIS	7°	
P12	75	M	Pathologic myopia	0.6	0.9	4°	6°
P13	70	M	Diabetic retinopathy	0.8	NPL	2°	
P14	54	W	Pathologic myopia	1	NPL	4°	

BCVA: best corrected visual acuity; PRL: preferred retinal locus; HM: hand motion CF: count fingers.

**Table 2 jpm-12-01709-t002:** Patients with peripheral vision loss characteristics.

ID	Age	Gender	Eye Disease	BCVA (LogMAR) Best Eye	BCVA (LogMAR) Worst Eye	BVF %
P1	70	W	Glaucoma	0.4	NPL	3.5
P2	75	M	Glaucoma	0.3	NPL	5.5
P3	52	M	Glaucoma	1.3	1.3	11.5
P4	38	W	Retinitis pigmentosa	0	0	23.5
P5	66	M	Diabetic retinopathy	0.7	0.7	22
P6	56	M	Glaucoma	0.5	HM	8.5
P7	69	M	Optic Atrophy	0.5	NPL	6
P8	59	W	Glaucoma	0.6	NPL	17.5

BCVA: best corrected visual acuity; BVF: peripheral binocular visual field; HM: hand motion NPL: no light perception.

**Table 3 jpm-12-01709-t003:** Data obtained from Sensory Organization Tests (SOT), Sensory Analyses (SA) and Motor Control Tests (MCT). CLV: central low vision; PLV: peripheral low vision; NC: normal control; SOT (1–6): trial condition; SOM: somatosensory contribution, VIS: visual contribution; VEST: vestibular contribution; PREF: visual preference MCT F: motor control test of medium (M) or large (L) translation in forward (F) or backward (B) directions. The columns on the right show the presence of significance between group comparisons (see also Figure 3 and Figure 4).

**SOT**	**CLV**	**PLV**	**NC**	**NC/CLV**	**NC/PLV**	**CLV/PLV**
1	87.835 ± 4.766	88.75 ± 3.228	91.3 ± 1.828	No	No	No
2	82.514 ± 10.039	82.175 ± 3.703	89 ± 1.943	No	No	No
3	75.442 ± 10.259	78.5 ± 5.65	88.5 ± 5.169	Yes	Yes	No
4	29.714 ± 12.338	35.237 ± 6.875	81.6 ± 3.977	Yes	Yes	No
5	50.971 ± 12.051	67.737 ± 7.466	75.2 ± 5.452	Yes	No	Yes
6	55.764 ± 8.645	62.325 ± 4.194	70.8 ± 4.96	Yes	Yes	No
CES	63.706 ± 6.227	68.641 ± 4.414	82.73 ± 1.846	Yes	Yes	No
**SA**	**CLV**	**PLV**	**NC**	**NC/CLV**	**NC/PLV**	**CLV/PLV**
SOM	0.937 ± 0.083	0.926 ± 0.040	0.975 ± 0.024	No	No	No
VIS	0.336 ± 0.134	0.396 ± 0.071	0.894 ± 0.049	Yes	Yes	No
VEST	0.584 ± 0.061	0.763 ± 0.084	0.823 ± 0.061	Yes	No	Yes
PREF	0.949 ± 0.049	0.974 ± 0.028	0.997 ± 0.032	Yes	No	No
**MCT F**	**CLV**	**PLV**	**NC**	**NC/CLV**	**NC/PLV**	**CLV/PLV**
M Dx	203.3 ± 6	141.7 ± 5.6	136.8 ± 5.9	Yes	No	Yes
L Dx	189.5 ± 7.3	123.6 ± 4.5	121.9 ± 3.1	Yes	No	Yes
M Sx	202.4 ± 5	157.4 ± 4.1	137.3 ± 4.2	Yes	Yes	Yes
L Sx	190.9 ± 5.7	125.1 ± 4	118.2 ± 8	Yes	No	Yes
**MCT B**	**CLV**	**PLV**	**NC**	**NC/CLV**	**NC/PLV**	**CLV/PLV**
M Dx	199.4 ± 5.6	136.4 ± 4.6	135.5 ± 3.2	Yes	No	Yes
L Dx	187.9 ± 6.5	117.6 ± 5.6	118.2 ± 4.4	Yes	No	Yes
M Sx	195.7 ± 8.5	146.5 ± 6.3	137.7 ± 4.7	Yes	Yes	Yes
L Sx	187.1 ± 6.3	115.5 ± 8.3	118.1 ± 5.2	Yes	No	Yes

## Data Availability

Not applicable.

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
