# Peer review of "Posturographic Analysis in Patients Affected by Central and Peripheral Visual Impairment"

_jpm, 2022, doi:10.3390/jpm12101709_

Round 1
Reviewer 1 Report
In this paper, Cadoni et al. performed a posturographic analysis on patients with central and peripheral visual impairment. The authors reported variations in postural stability between CLV and PLV patients. The overall study design, methods, and figures are well done. The interpretation is consistent with the findings. However, some experimental concerns reduce confidence in the strength of the conclusions and the potential impact of the study on the field. Below are my concerns:
My first concern is the relatively small sample size. Will it be possible for the authors to increase the sample size? Also, authors should check if samples are age-matched (NC vs CLV and PLV).
There are many factors that can influence postural control, including age, lower limb muscle strength, and physical activity. In general, increasing age, limb muscle weakness, and less physical activity negatively affect postural stability. The authors did not mention these factors in the manuscript. The authors should include these points in the discussion part.
The CLV group has 14 patients’ data and the age range of this group is 38-82 years. Authors can analyze this group based on age (compare the postural stability between 38-58 vs 68-above CLV patients) and check if there is any difference in postural stability within the group.
The manuscript has many typos and grammatical errors, I have listed some of them:
Line 13> informations >> information.
Line 27 > seems>> seem.
Line 43 > affect >> affects.
Line 63-66> ‘Visual field gradually reduces……navigate in walking’.>> This sentence is not correctly constructed, consider revising it.
Line 68 > diabetic retinopathy associated with >> is associated.
Line 70 > such us >> such as.
Finally, A more tightly worded manuscript with the above-suggested corrections will make the paper a good read.
Author Response
Thank you for your appropriate comments on our manuscript. Point 1. My first concern is the relatively small sample size. Would it be possible for the authors to increase the sample size? Furthermore, authors should check if samples are age-matched (NC vs CLV and PLV). Response 1.We agree with you relatively to the limitations about our sample size. However, we would underline that we enrolled into this study all possible subjects according to the inclusion criteria who underwent to the clinic.Thus, unfortunately, even if advisable, it was not possible to improve our sample size. Anyway, we preliminarily evaluated the statistical appropriateness of our study according to the normative data of Neurocom and to many data of literature (Di Girolamo et al. 2009; Calo' et al. 2009; Neil T Shepard et al. 2010). Therefore, the enrollement of 8 patients for each group appeared sufficient to obtain a significant result (power size 95%, Zpwr value =1,64, p-value =0,01, significancy =99%, Zcrit value= 2,58). Certainly, our data need to be confirmed by further studies but we hope that our results could be useful to promote further research. Point 2. There are many factors that can influence postural control, including age, lower limb muscle strength, and physical activity. In general, increasing age, limb muscle weakness, and less physical activity negatively affect postural stability. The authors did not mention these factors in the manuscript. The authors should include these points in the discussion part. Response 2. Thank you for your suggestion. We updated the main text, pointing out the small sample size limitation and adding the following statement: "Postural control is a complex process which can be influenced by many factors like aging, physical conditions, cognitive functions and many age-related diseases. Particularly in elderly population cognitive or physical inactivity could be associated with a lower balance control leading to an higher risk of fall. The sample size of our study is not adequate for a stratification of data relating to these variables. To obtain a sample as homogeneous as possible, our study protocol excluded that patients underwent any kind of major surgery or suffering from serious pathologies (neurological, orthopedic, physiatric, vestibular or Otologyc disease or serious other comorbidities). We enrolled subjects with a healthy lifestyle, adequate personal autonomy and age-appropriate physical activity, as many authors underlined that even leisure physical activity is very effective for balance control amelioration and fall prevention. Point 3. The CLV group has 14 patients data and the age range of this group is 38-82 years. Authors can analyze this group based on age (compare the postural stability between 38-58 vs 68-above CLV patients) and check if there is any difference in postural stability within the group. Response 3. We checked our statistical analysis. As regard to CLV group, in younger subjects (38-58) ES score reached the value of 64,27 (+- 5,50), in 68> patients, ES score was 62,95 (+-3,95). No significant difference was detected between these groups (p=0,71), probably due to both patient selection and asymmetric age distribution within the 38-82 age range. Sample groups were age-matched. If requested, graphs and data may be added to supplementary section. Point 4. The manuscript has many typos and grammatical errors, I have listed some of them Text was corrected according to your suggestions, sentence in line 63-66 was changed.Reviewer 2 Report
Thanks the authors for conducting this cross medical subspecialty study, it is interesting on the sensory system of human.
The ophthalmology examinations are detailed and appropriate, with Best Corrected Visual Acuity (BCVA) assessed through Early Treatment Diabetic 100 Retinopathy Study charts, Reading Acuity (RA) assessed by the Minnesota Reading test (MNRead) charts at 25 cm using a +4.00 sph (1X) reading lenses in addition to the distance refractive adjustment, Contrast sensitivity evaluated through Pelli Robson charts at a distance of 1 m, Fixation stability assessed using the Nidek Technologies MP-1 microperimeter, Peripheral Binocular Visual Field (BVF) evaluated with Humphrey Field Analyzer 119 II (Zeiss) automated static perimeter.
However, the low number of subjects in each comparison group might affect the significance of results. Sample size calculation should be demonstrated and presented in Methods in order to convince audience. Larger sample size is actually needed for visual field related study.
Author Response
Thank you for your appropriate comments to our manuscript. We agree with you on the limitations about our sample size. In our study we enrolled all possible subjects according to the inclusion criteria which underwent to the clinic. We preliminarily evaluated the statistical appropriateness of our design, taking into account Neurocom normative data and some published articles (Di Girolamo et al. 2009; Calo’ te al. 2009; Neil T Shepard te al. 2010). Therefore, the enrollement of 8 patients for each group appeared sufficient to obtain a significant result (power size 95%, Zpwr value =1,64, p-value =0,01, significancy =99%, Zcrit value= 2,58). Certainly, our data need to be confirmed by further studies but we hope that our results could be useful to promote further research. We updated the main text adding the following statement in methods section: "Taking into account Neurocom international normative data on Equitest Equilibrium Score in healthy adult population and considering an expected result sufficient to obtain a pathological Equilibrium Score, our sample size, altought limited, was adequate (study power of 95%, Zpwr value = 1.64, p-value = 0 , 01, significance = 99%, Zcrit value = 2,58).